# Multiple Transitions between Y Chromosome and Autosome in Tago’s Brown Frog Species Complex

**DOI:** 10.3390/genes15030300

**Published:** 2024-02-26

**Authors:** Ikuo Miura, Foyez Shams, Jun’ichi Ohki, Masataka Tagami, Hiroyuki Fujita, Chiao Kuwana, Chiyo Nanba, Takanori Matsuo, Mitsuaki Ogata, Shuuji Mawaribuchi, Norio Shimizu, Tariq Ezaz

**Affiliations:** 1Amphibian Research Center, Hiroshima University, Higashi-Hiroshima 739-8526, Japan; ahorn27g10@gmail.com (C.K.); nnbcy@hiroshima-u.ac.jp (C.N.); 2Institute for Applied Ecology, Centre for Conservation Ecology and Genomics, Faculty of Science and Technology, University of Canberra, Canberra, ACT 2617, Australia; foyez.shams@canberra.edu.au (F.S.); tariq.ezaz@canberra.edu.au (T.E.); 3Natural History Museum and Institute, Chiba 260-8682, Japan; ranatagoi@gmail.com; 4Gifu World Freshwater Aquarium, Kakamigahara, Gifu 501-6021, Japan; tagaminmin@gmail.com; 5Saitama Museum of Rivers, Yorii-Machi, Oosato-Gun, Saitama 369-1217, Japan; zx900a14@yahoo.co.jp; 6Department of Preschool Education, Nagasaki Women’s Junior College, Nagasaki 850-0823, Japan; gekkonishi@yahoo.co.jp; 7Preservation and Research Center, City of Yokohama, Yokohama 241-0804, Japan; zvp06246@nifty.com; 8National Institute of Advanced Industrial Science and Technology (AIST), Tsukuba 305-8568, Japan; mawaribuchi.s@aist.go.jp; 9Hiroshima University Museum, Higashi-Hiroshima 739-8524, Japan; norios@hiroshima-u.ac.jp

**Keywords:** sex chromosome turnover, hybridization, introgression, speciation

## Abstract

Sex chromosome turnover is the transition between sex chromosomes and autosomes. Although many cases have been reported in poikilothermic vertebrates, their evolutionary causes and genetic mechanisms remain unclear. In this study, we report multiple transitions between the Y chromosome and autosome in the Japanese Tago’s brown frog complex. Using chromosome banding and molecular analyses (sex-linked and autosomal single nucleotide polymorphisms, SNPs, from the nuclear genome), we investigated the frogs of geographic populations ranging from northern to southern Japan of two species, *Rana tagoi* and *Rana sakuraii* (2n = 26). Particularly, the Chiba populations of East Japan and Akita populations of North Japan *in R. tagoi* have been, for the first time, investigated here. As a result, we identified three different sex chromosomes, namely chromosomes 3, 7, and 13, in the populations of the two species. Furthermore, we found that the transition between the Y chromosome (chromosome 7) and autosome was repeated through hybridization between two or three different populations belonging to the two species, followed by restricted chromosome introgression. These dynamic sex chromosome turnovers represent the first such findings in vertebrates and imply that speciation associated with inter- or intraspecific hybridization plays an important role in sex chromosome turnover in frogs.

## 1. Introduction

Sex chromosome turnover refers to the transition that occurs between sex chromosomes and autosomes, and it recurs as long as they remain in a homomorphic state [1,2,3,4,5,6]. When the sex chromosomes evolve to become heteromorphic, they are evolutionarily conserved in association with Y or W chromosome degeneration over a period of a billion years, as observed in therian mammals, birds, and sharks [7,8,9]. However, if certain populations conserving the state of homomorphy remain within the species or in sister species, homomorphy can be resurrected through hybridization, as observed in the Japanese frog *Glandirana rugosa* [10] and predicted in fish and amphibians [11]. These represent the turnover of heteromorphic sex chromosomes. To date, several evolutionary reasons for sex chromosome turnover have been proposed, including sex-ratio bias, genetic drift, sex-antagonistic genes, and genetic load accumulated on the Y or W chromosome [12]. Additionally, sex chromosome turnover has often been discussed in terms of speciation [13,14,15]. From the perspective of speciation, Japanese frogs are interesting because sex chromosome turnover frequently occurs and they experience speciation induced by geographic movements such as the elevation of sea levels, isolation by mountains, and/or fusion and fission between islands and continents through land bridges at glacial ages [16,17]. Thus, frogs living on the Japanese Islands are suitable models for studying the relationship between sex chromosome turnover and speciation.

Japanese Tago’s brown frogs are commonly found in mountainous regions and have undergone rapid speciation, as evidenced by various studies [18,19]. Originally, *Rana tagoi*, which is distributed across all Japanese islands, except Hokkaido (northern large Island), was described as a single species [20]. In 1990, the stream-dwelling species *R. sakuraii* was identified [21]. Since then, three additional species have been described [22,23,24] and the existence of several cryptic species has been suggested [18,19].

The two Tago’s brown frog species, *R. tagoi* and *R. sakuraii*, possess 26 diploid chromosomes and exhibit highly conserved karyotypes, similar to other Ranid frogs [1,2]. In 2021, Kuwana et al. [25] confirmed that the chromosome 7 of *R. sakuraii* (Tokyo) was the heteromorphic XX/XY sex chromosome (chromosome 8, as described in the original paper by [26]). Additionally, the heteromorphic sex chromosome of *R. tagoi* from Tokyo was identified as chromosome 7 (chromosome 8 as described by [26]). Kuwana and colleagues also discovered that the homologous chromosome 7 of *R. sakuraii* from the West Japan population was homomorphic in both sexes and did not show any morphological differences between the sexes. In contrast, the Y chromosome of *R. sakuraii* from East Japan is morphologically similar to that of *R. tagoi* from Tokyo and nearby populations. The researchers proposed that the Y chromosomes of *R. sakuraii* of Tokyo were introduced from the sister species, *R. tagoi*, making this frog species complex an ideal model for studying sex chromosome turnover and speciation [25].

The present study aimed to validate the Y chromosome introgression between the two sister frog species by examining sex-linked markers of single nucleotide polymorphisms (SNPs) and extending the research to other populations in the North and South. Surprisingly, our findings revealed repetitive transitions between the Y chromosome and autosome because of inter- or intraspecific hybridization.

## 2. Materials and Methods

### 2.1. Frogs

The collection locations and number of Tago’s brown frogs *R. tagoi* and *R. sakuraii* used for this study are presented in Figure 1 and Appendix A. The frogs from Chiba populations of East Japan and Akita populations of North Japan in *R. tagoi* were newly investigated in addition to those used in our previous study [25]. Chiba population was chosen because it is suggested to be ancestral lineage of Tokyo populations in *R. tagoi* [18], while Akita population was chosen because we preliminarily found unusual and different karyotype from other populations. In comparison to one or two populations collected from South Japan of *R. tagoi* and West Japan of *R. sakuraii*, many more populations of the two species were collected from East Japan in order to deeply investigate the relationship between speciation and sex chromosome evolution in the region. Sib-ship offspring were obtained from fertilized eggs by spontaneously mating at the facility of Hiroshima University between a pair of one male and one female collected from each population, or egg masses directly were collected from each population and were reared at the facility of Hiroshima University. The embryos were reared until sexual maturation and used for development of SNPs and for chromosome analyses. Animal care and experimental procedures were conducted with the approval of the Committee for Ethics in Animal Experimentation at Hiroshima University (permit no.: G22-4), and the approval date was 24 October 2022.

### 2.2. Mitochondrial Sequence Analysis

Following the previous phylogenetic studies on *R. tagoi* and *R. sakuraii* [18,19], to amplify the fragment of mitochondrial NADH dehydrogenase, subunit 1 (ND1) and 16S rRNA genes, PCR was performed using Sapphire AmpFast PCR master mix (TaKaRa, Japan) as follows: 1 μL of DNA was amplified in 25 μL reaction volume containing 12.5 μL 2-fold Premix including Taq polymerase and dNTP and 0.5 μL of each of the 12.5 μM primers at 94 °C for 40 s, 60 °C for 20 s, and 72 °C for 20 s for 35 cycles using Smart cycler (TaKaRa). The primers used were the same as those of our previous study [25]. The amplified fragments were purified using FastGeneTM Gel/PCR Extraction Kit (Nippon Genetics, Tokyo, Japan) and the nucleotide sequence was determined using an ABI PRISM 3130xl genetic analyzer (Applied Biosystems, Waltham, MA, USA) according to the manufacturer’s instruction. The phylogenetic tree of nucleotide sequences was constructed as described previously [27]. Alignment was performed using MUSCLE [28]. The best-fit model of nucleotide substitution (TIM2 + I + G4) was selected by Modeltest-NG 0.1.6 [29]. Maximum likelihood phylogenetic tree with 100 bootstrap replicates was constructed using RAxML-NG 1.0.2 [30]. The phylogenetic tree was drawn using FigTree (http://tree.bio.ed.ac.uk/software/figtree/, accessed on 20 February 2023). Sequence data from this article deposited with the DDBJ data libraries and from other papers are listed in Appendix A.

### 2.3. Chromosome Preparation and Banding Techniques

Mitotic metaphase chromosomes were obtained from a blood cell culture [25]. Late-replication (LR-) banding and C-banding techniques were generally used, as previously described [25,31,32]. The number of specimens used for chromosome analyses are shown in Appendix A. We photographed 10–30 good C- or LR-banded metaphases from those of each frog that we observed, and presented the best ones in the figures.

### 2.4. Genotyping by Sequencing Using DArTseq

To generate genome-wide single-nucleotide polymorphism (SNP) and SilicoDArT (presence/absence [PA]) markers, we used genotyping-by-sequencing (GBS) technology developed by Diversity Arrays Technology (DArT) Pty. Ltd. (University of Canberra, Canberra, ACT, Australia). Total genomic DNA was extracted from finger clip tissues of the frogs using NucleoSpin Tissue (TAKARA, Kyoto, Japan), according to the manufacturer’s instruction, and was used for GBS [33,34,35].

### 2.5. Identification of Sex-Linked Markers

We combined previously published pipelines to identify both SNP and PA sex-linked loci in this study [2,34,36]. To incorporate potential recombination between sex chromosomes and possible sex reversal within these species and minor sequencing errors within the data set, we applied an 80% concordance threshold to isolate sex-linked loci [37]. First, we filtered out all loci (both SNPs and SilicoDArT) that were below the 80% call ratio. The remaining loci were tested for the presence of both male heterogametic sex determination (XX/XY) and female heterogametic sex determination (ZZ/ZW) systems. To filter out SNPs and PA that support the filtering criteria but are not true sex-linked loci, we performed a false positive test across all identified sex-linked loci by calculating the proportion of homozygous alleles, as described by [2].

### 2.6. Alignment of Sex-Linked Genes to Rana temporaria Reference Genome

To determine the approximate locations of sex-linked markers isolated from two *Rana* species, we aligned sequences to the closest available reference genome using makeblastdb (NCBI blast+), i.e., that of the European common frog *Rana temporaria*. This assembly (available at https://github.com/DanJeffries/Rana-temporaria-genome-assembly/wiki, accessed on 24 May 2021) consists of PacBio sequencing, is scaffolded using optical mapping data, and is anchored to chromosomes using an existing linkage map [2].

### 2.7. Principal Component, Structure, and Gene Tree Analysis

For the population genetic structure analysis using autosomal loci, we filtered out loci that contained more than 1% null allele (99% call ratio), had <100% reproducibility, and were not sex-linked. Principal component analysis (PCA) was performed using the “dartR” (v2.7.2) package [38] on statistical computing software R (v4.1.1) [39]. The results were plotted using the “ggplot2” (v3.3.5) package [40]. The genetic structure analysis was performed using software “Structure” version 2.3.4 [41]. The analysis was performed using an admixture model with 10,000 burnin and 10,000 MCMC assuming 10 populations. Each iteration was run 10 times to identify appropriate K value among the data set. The deltaK was calculated using online platform “Structure Harvester” [42]. Identification of best-fit models and unrooted gene trees were created using MEGA 11 software [43].

## 3. Results

### 3.1. Sex Chromosomes of R. tagoi from Chiba (East Japan) and Akita (North Japan) Populations

To identify the sex chromosomes morphologically, we investigated the chromosomes of Chiba populations of *R. tagoi* in East Japan and Akita populations in North Japan (Figure 1). Based on mitochondrial DNA sequences, Chiba populations took the ancestral position of *R. tagoi* and *R. sakuraii* in East Japan, while Akita populations were divided into two different lineages, Akita-B, belonging to the clade of the Tohoku district with other populations, including another sister species *R. Kyoto*, in North Japan, and Akita-A, which occupied a unique position of an ancestral lineage in the Tohoku clade (Appendix A).

Chromosome 7 of the Chiba population was found to be subtelocentric in both sexes, but with a slightly longer short arm on one homologue compared to that of the other in males. On the other hand, chromosome 7 was homomorphic for the former homologue in females (Figure 2, Appendix A). These findings indicated the presence of an XX-XY sex chromosome. In contrast, chromosome 7 in the Akita-B populations of North Japan was metacentric in both sexes (Appendix A), similar to the karyotypes of South Japan populations and *R. sakuraii* from West Japan populations [25], and showed no differences between sexes. Chromosome 7 of Akita-A was unique, subtelocentric in both sexes, morphologically similar to the Y chromosomes of Chiba populations, and showed no morphological sex chromosome differentiation (Figure 3, Appendix A). Unexpectedly, in two males of the Akita-B population, chromosome 7 was heteromorphic, metacentric, and subtelocentric, similar to the X and Y chromosome 7 of *R. sakuraii* from East Japan (designated Akita-B’ in Appendix A).

### 3.2. Identification of Sex Chromosomes Based on the Sex-Linked SNP Markers

To confirm or identify the sex chromosomes of *R. tagoi* and *R. sakuraii*, we isolated sex-linked markers of SNP from sib-ships of Chiba (East Japan), Akita-A and -B’ (North Japan), Nagasaki (South Japan) of *R. tagoi*, and Saitama (East Japan) and Gifu (West Japan) of *R. sakuraii* (Table 1; Appendix A). The sex-linked marker sequences were then aligned to the reference genome of the European brown frog *R. temporaria* using BLASTn 2.2.26+, and sex-linked markers aligned to a single chromosome were selected.

Then, 28 (93.1%), 78 (81.3%), and 57 (89.1%) of the 30, 96, and 64 sex-linked markers of Chiba and Akita-B of *R. tagoi* and *R. sakuraii* (Saitama), respectively, were aligned to chromosome 9 (Figure 4a–c), which corresponds to chromosome 7 of *R. tagoi* and *R. sakuraii* [1,2,27,44]. In contrast, 7 (87.5%), 3 (60%), and 5 (83.3%) of the 8, 5, and 6 sex-linked markers of Nagasaki and Akita-A of *R. tagoi* and *R. sakuraii* (Gifu), respectively, were aligned to chromosomes 13, 4, and 13, respectively (Figure 4d–f), which correspond to chromosomes 13, 3, and 13 of *R. tagoi* and *R. sakuraii*, respectively [1,2,27,44].

### 3.3. The Origins of Y Chromosomes and Autosome 7

To elucidate the evolutionary origins of the Y chromosomes (No. 7) identified in *R. tagoi* and *R. sakuraii*, we analyzed the STRUCTURE, principal component, and ML gene tree based on 219 sex-linked markers identified in the two species (Figure 5). The STRUCTURE map revealed that *R. sakuraii* males from East Japan shared half of the clusters with females (indicated in red), while the other half with *R. tagoi* individuals from Tokyo (indicated in pink) on the K = 5 map (Figure 5a). This finding implies that the Y chromosome of *R. sakuraii* in East Japan was introduced from the Tokyo population of *R. tagoi* (curved arrow ①). Most clusters of *R. tagoi* from Tokyo are shown in green on the K = 4 map, which is similar to the Akita-A population. This suggests that the Y chromosomes of Tokyo were derived from autosome 7 of Akita-A (as indicated by the curved arrow ② in the K = 4 map). Similarly, half of the clusters of one male Akita-B, indicated by the small green arrow in the K = 4 map, which has a heteromorphic pair of chromosome 7, are shown in green as those of Akita-A (indicated by the curved arrow ③ in the K = 4 map). This suggests that autosome 7 of Akita-A was introduced into Akita-B and evolved into the Y chromosome. Finally, individuals of Akita-A are shown in green and blue in the K = 3 map, suggesting that autosome 7 was derived from the Y chromosomes of Chiba populations, shown in green (indicated by the curved arrow ④ in the K = 3 map of Figure 5a). These chromosome 7 introgressions were supported by an additional PCA and gene tree (Figure 5b,c).

### 3.4. The Population Dynamics of the Two Species

In order to confirm whether inter- or intraspecific hybridization resulted in chromosome introgression, we analyzed the STRUCTURE, principal component, and ML gene tree based on 2164 autosomal SNP shared by the two species (Figure 6). In the STRUCTURE map shown in Figure 6a, the Akita-A clusters of *R. tagoi* comprised three different clusters of *R. sakuraii* from East Japan, Chiba, and Ehime/Nagasaki of *R. tagoi* (as indicated by the dotted line box A at K = 3 and 4). In contrast, Akita-B was found to be comprised of two different clusters of the Akita-A and Ehime/Nagasaki populations (as indicated by the dotted line box B in K = 5). Additionally, the Tokyo populations of *R. tagoi* comprised three distinct clusters, comprising Akita-A and the Ehime/Nagasaki populations of *R. tagoi* and *R. sakuraii* from East Japan (indicated by the dotted line box C in K = 5). These results suggest that inter- or intraspecific hybridization occurred in these populations, which was further supported by the findings of PCA and gene tree analysis (Figure 6b,c).

## 4. Discussion

### 4.1. Sex Chromosomes in Tago’s Brown Frogs

The sex chromosomes identified in *R. tagoi* and *R. sakuraii* are summarized in Table 2. The findings of this study, based on sex-linked SNP, show that the sex chromosomes of *R. sakuraii* from East Japan (Saitama) are chromosome 7 (Figure 4), as reported earlier [25], and those of West Japan are chromosome 13. We also confirmed that the sex chromosomes of *R. tagoi* from South Japan (Nagasaki) are represented by chromosome 13 (Figure 4 and Appendix A), coinciding with the sex-linked chromosome heteromorphy (ref. [25] and Appendix A). The Chiba populations of *R. tagoi* are geographically located near Tokyo and phylogenetically assume an ancestral position in the clade based on the mitochondrial gene tree (refs. [18,19,25]; Appendix A in this study). Further, we found that the frogs have a heteromorphic pair of subtelocentric chromosome 7 in males, as seen in the Tokyo population [26]. On the other hand, we investigated the population of the Akita Prefecture in North Japan for the first time (Higashi-naruse village). Akita-B populations had metacentric chromosome 7 in both sexes, but two males (Akita-B’) had a heteromorphic pair of chromosome 7, metacentric and subtelocentric (referred to as Akita-B’), which were confirmed to be sex chromosomes, as in *R. sakuraii* from East Japan (Figure 4). The Akita-A populations were unique and had subtelocentric chromosomes in both sexes, and the sex chromosome was suggested to be chromosome 3, but not chromosomes 7 or 13, based on SNP (Figure 4). A more comprehensive analysis is essential to prove the existence of sex chromosome 3, given the limited number of sex-linked markers that have been identified. However, it is evident that no sex-linked markers were isolated from chromosome 7 or 13 in the Akita-A populations. In summary, our findings revealed the presence of three distinct sex chromosomes, namely chromosomes 3, 7, and 13, in the two species (Table 2), suggesting that the occurrence of sex chromosome turnover is related to the transition between Y chromosome and autosome 7.

### 4.2. Transitions between Y Chromosome and Autosome

The findings of the present study prove that the repetition of chromosome 7 transition between the Y chromosome and autosome is associated with population differentiation through inter- or intraspecific hybridization (Figure 7). Specifically, the Y chromosome (Y_1_) of the Chiba population of *R. tagoi* was introduced and changed to autosome 7 in the Akita-A population, and was subsequently reintroduced to the Y chromosomes in the Tokyo (Y_2_) and Akita-B’ (Y_3_) populations by chromosome introgression. More recently, the Y chromosome (Y_2_) of the Tokyo population was introduced into *R. sakuraii* from East Japan and remains on the Y chromosome. A return transition from the X chromosome to the autosome has also been identified in *Drosophila* fruit flies [45]. A surprising finding in Tago’s brown species is the recurrence of the transition between the Y chromosome and the autosome. To the best of our knowledge, this is the first report on the evolution of sex chromosomes. This transition is associated with restricted chromosomal introgression through inter-or intraspecific hybridization. This form of sex chromosome evolution via chromosome introgression through interspecific hybridization has been observed in fish and frogs. The Y chromosome of the nine-stickleback fish (*Pungitius pungitius*) evolved from homologous chromosome 12 (probably the Z chromosome), which was introduced from the closely related species *Pungitius sinensis* [46]. In the Japanese frog *G. rugosa*, the X and W chromosomes evolved from autosome 7, which was introduced from the sister species *G. reliquia* (the former East Japan population) through hybridization [1,44]. The differentiation of sex chromosomes typically begins from a homomorphic pair, with the Y or W chromosome then accepting chromosomal rearrangements and structural changes, such as the amplification of heterochromatin and transposons, and proceeding toward differentiation and degeneration [7]. However, the introduction of a differently shaped homologous chromosome from a different species can quickly constitute a heteromorphic pair, including a large non-recombining region, as seen in frogs, fish, and even birds (the second sex chromosome in [47]). This phenomenon can be considered as a shortcut to the evolution of the sex chromosome.

### 4.3. Hybridization and Sex Chromosome Turnover

Sex chromosomes in most frog species are homomorphic in both sexes [48,49]. These chromosomes exhibit frequent turnover between species and even in geographic populations of the same species [1,2]. Various theories have been proposed regarding the evolutionary reasons associated with sex chromosome turnover, including sex-ratio distortion [50,51], genetic drift [52], sex-antagonistic gene selection [53,54], and the genetic load caused by the accumulation of deleterious mutations on Y or W chromosomes [55]. However, the third possibility, sex-antagonistic genes, may not be relevant to frogs as they do not ride on sex chromosomes [12,56,57,58,59]. Genetic drift could be the explanation, as sex chromosome turnover often occurs during speciation or geographic population differentiation in Japanese and European frogs, such as *Pelophylax porosus* [16], *R. japonica* [17], and some species of *Pelophylax* and *Hyla* [2,14]. The fourth possibility, the genetic load, would be elucidated in the future when whole-genome analysis would be completed for most frog species that have undergone sex chromosome turnover. Sex-ratio distortion may also be a plausible reason for the evolution of novel XY and ZW sex chromosomes through hybridization between *G. rugosa* and its sister species *G. reliquia* [1,44]. Hybridization between two species with sex chromosomes 1 and 3 resulted in the creation of new populations with heteromorphic sex chromosome 7 [44], in which the sex-ratio was assumed to be distorted based on crossing experiments under captive breeding [50]. The results of this study suggest that inter- or intraspecific hybridization played a role in sex chromosome turnover. In the Tago’s brown frogs, three different sex chromosomes, 3, 7, and 13, underwent sex chromosome turnover. These incidents were unique in that two or three populations of the two species participated in a single hybridization event (Figure 8). It is speculated that hybridization between populations with dissimilar genetic backgrounds may have disrupted the native genetic mechanisms of sex determination in each population, followed by sex-ratio skewing, which may have necessitated the evolution of a new sex-determining mechanism driven by another male-determining gene originally located on one of the autosomes belonging to the fixed members of six potential sex chromosomes [1]. In addition, chromosome 13 was added to the members in this study (Appendix A).

## 5. Conclusions

We identified three distinct sex chromosomes, namely chromosome 3, 7, and 13, in the geographic populations of the Tago’s brown frog species *R. tagoi* and *R. sakuraii* (2n = 26), which are distributed from North to South Japan. In particular, chromosome 7 has been found to repeat a forward or backward transition between the Y chromosome and autosome as a result of hybridization between two or three populations belonging to the two species. The process of inter-or intraspecific hybridization followed by the introduction of a restricted chromosome from a different population or species may have played a significant role in the sex chromosome turnover.

## Figures and Tables

**Figure 1 genes-15-00300-f001:**
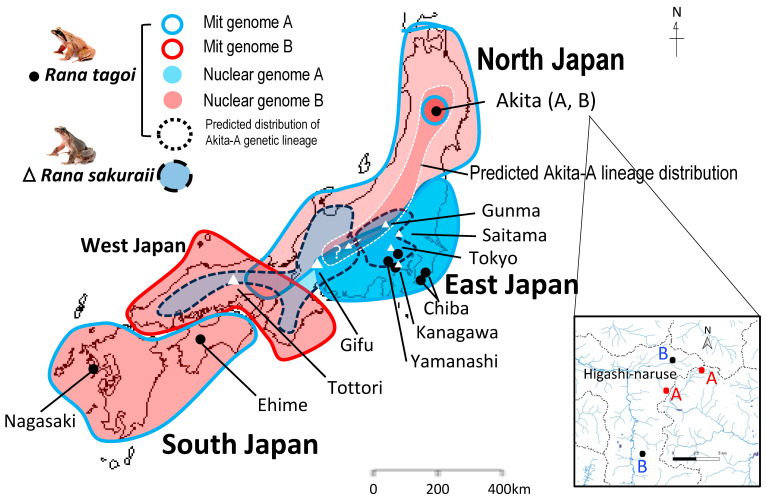
Japan map showing the locations of geographic populations of *R. tagoi* and *R. sakuraii*. Based on previous studies [15,16], the two major types, A and B, of mitochondrial and nuclear genomes are shown in blue and red, respectively. The area surrounded by the small white dotted line indicates the predicted distribution of the Akita-A (Higashi-naruse) genetic lineage based on published data and other information. The magnified map showing the detailed frog collection stations from Akita-A and -B (Higashi-naruse village) in North Japan is shown at the bottom right.

**Figure 2 genes-15-00300-f002:**
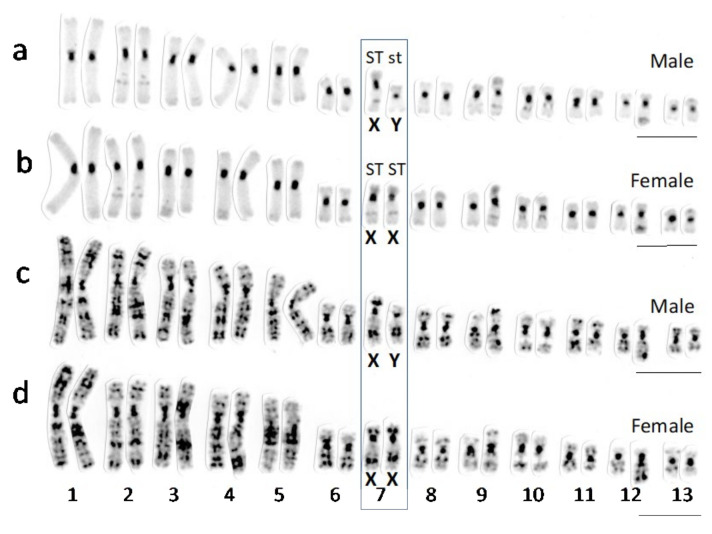
Karyotypes of *R. tagoi* from the Chiba population (Ichihara) of East Japan. C-banded (**a**,**b**) and late-replication-banded chromosomes (**c**,**d**). Chromosome 7 (boxed) is subtelocentric and heteromorphic (ST st) in males and homomorphic (ST ST) in females. The short arm of chromosome X (ST) is slightly longer than that of chromosome Y (st). Bar, 10 μm.

**Figure 3 genes-15-00300-f003:**
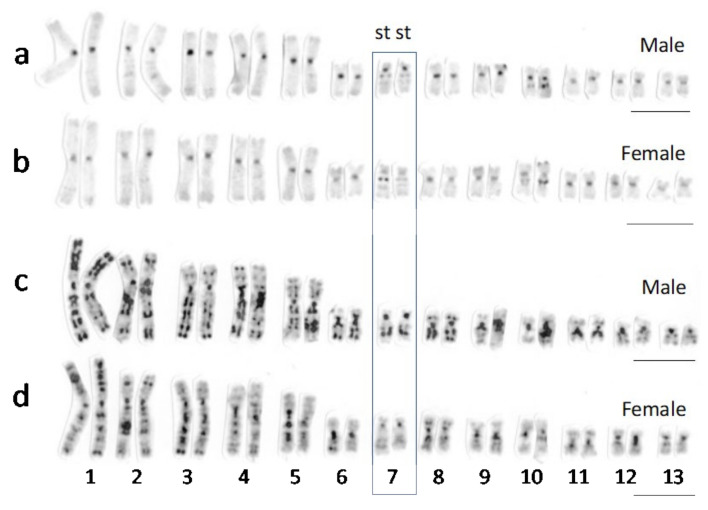
Karyotypes of *R. tagoi* from the Akita-A population (Higashi-naruse) of North Japan. C-banded (**a**,**b**) and late-replication-banded chromosomes (**c**,**d**). Chromosome 7 (boxed) is subtelocentric (st) and homomorphic in both the sexes. Bar, 10 μm.

**Figure 4 genes-15-00300-f004:**
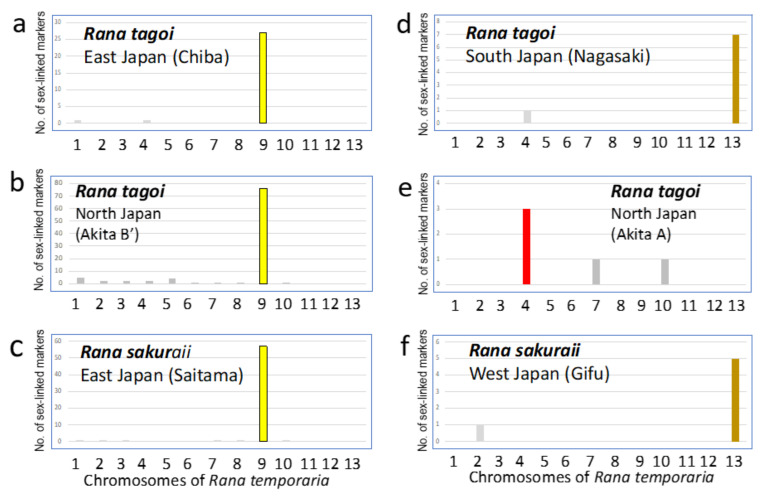
Graphs showing number of sex-linked SNP markers isolated from four populations. Chiba, Nagasaki, Akita-B’ and A of *R. tagoi* (**a**, **d**, **b,** and **e**, respectively), and two (Saitama and Gifu) of *R. sakuraii* (**c**,**f**). The markers were aligned to genomic sequences of chromosomes 1–13 of European brown frog *R. temporaria*. The bars showing the maximum number of sex-linked SNPs aligned to a single *R. temporaria* chromosome in 13 complements are colored and the others are in gray.

**Figure 5 genes-15-00300-f005:**
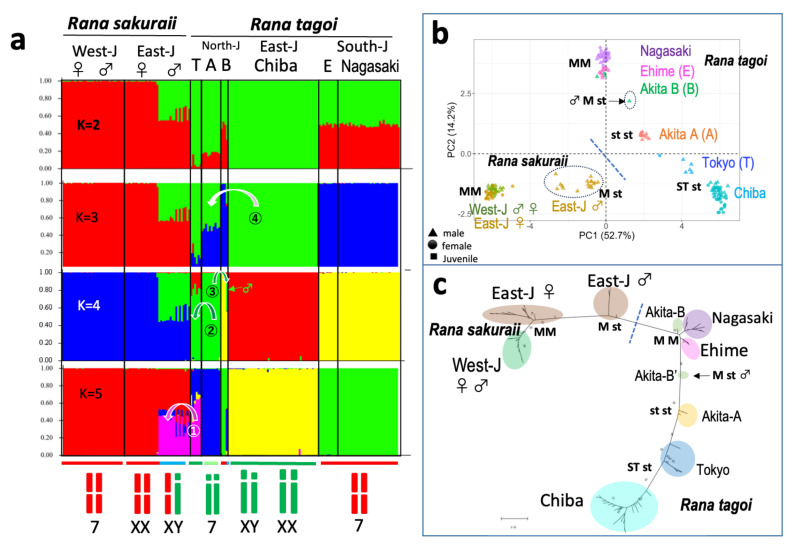
STRUCTURE (**a**), principal component analysis (**b**), and maximum likelihood gene tree (**c**) based on 219 sex-linked SNP shared by *R. tagoi* and *R. sakuraii*. Curved arrows in (**a**) indicate introgression of Y chromosome 7 or autosome 7 between the geographic populations of *R. tagoi* or from *R. tagoi* to *R. sakuraii*. Autosome 7 and sex chromosome 7 are shown at the bottom, under the colored horizontal bars (M/M in both sexes is in red, M/st in male is in blue, st st in both sexes is in pale green, and ST st in male is in green). The small green arrow indicates a male-bearing heteromorphic sex chromosome 7 in the Akita-B population (Akita-B’ designated in the text). Dotted short bars separate the two species in (**b**,**c**). The morphology of chromosome 7 is indicated by M (metacentric), ST (subtelocentric with a longer short arm), and st (subtelocentric with a shorter short arm). Males bearing heteromorphic X and Y chromosomes in (**b**) are circled with dotted lines. T, A, B, and E in (**a**) indicate Tokyo with Kanagawa and Yamanashi, Akita-A, Akita-B, and Ehime populations, respectively. Delta K was highest (737.97) in the K = 2 map.

**Figure 6 genes-15-00300-f006:**
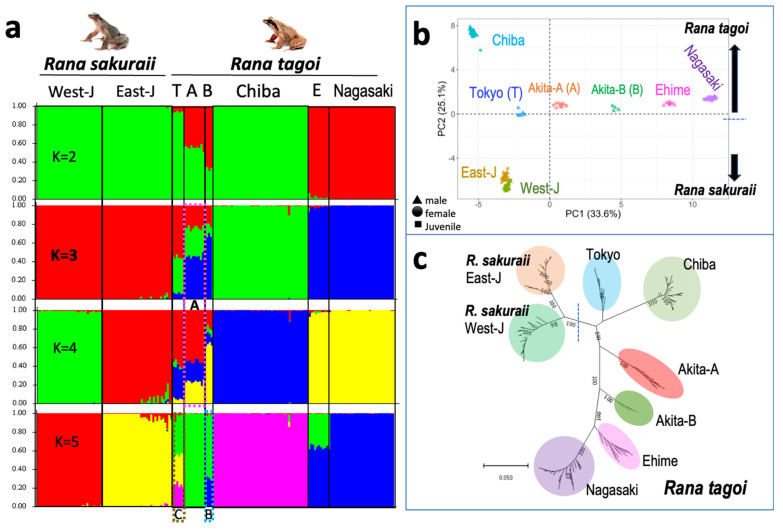
STRUCTURE (**a**), principal component analysis (**b**), and maximum likelihood gene tree (**c**) based on 2164 autosomal SNP shared by *R. tagoi* and *R. sakuraii*. Mixed clusters comprising two or three populations are indicated in boxes A, B, and C with colored dotted lines in pink, blue and brown, respectively. T, A, B, and E in (**a**) indicate Tokyo with Kanagawa and Yamanashi, Akita-A, Akita-B, and Ehime populations, respectively. Delta K was highest (998.34) in the K = 3 map.

**Figure 7 genes-15-00300-f007:**
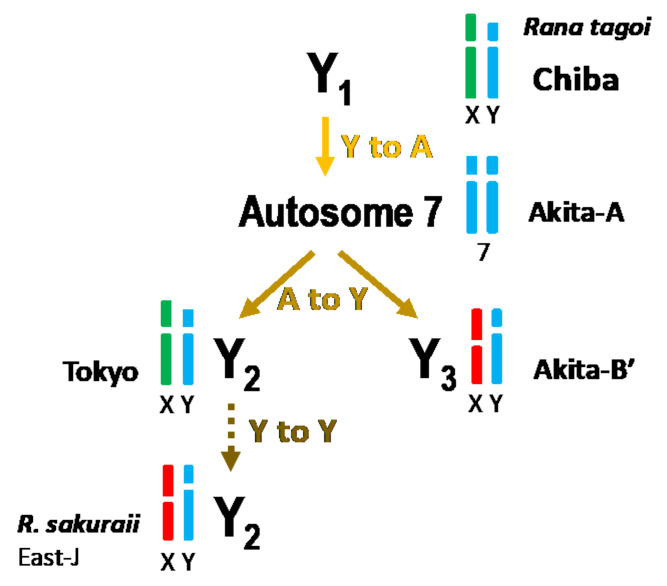
Transitions between the Y chromosome and autosome 7 identified in *R. tagoi* and *R. sakuraii*.

**Figure 8 genes-15-00300-f008:**
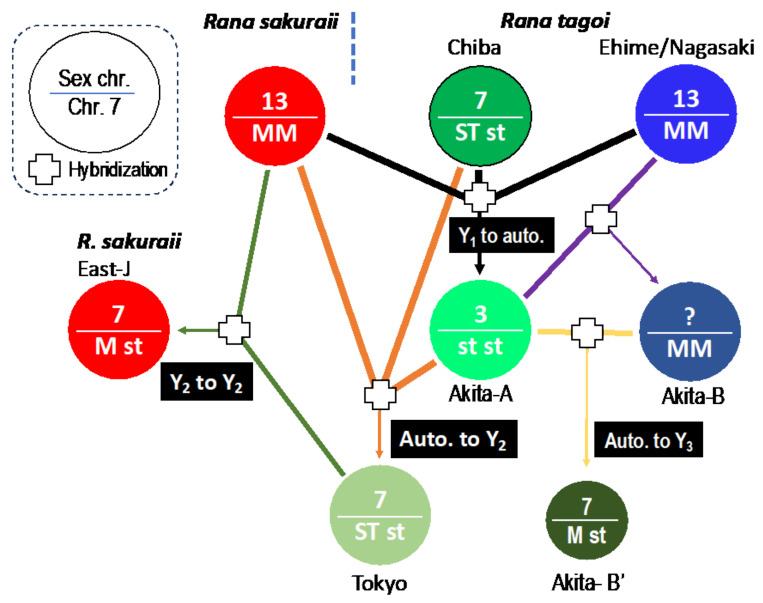
Sex chromosome turnovers associated with intra- or interspecific hybridizations in *R. tagoi* and *R. sakuraii*. Each of the populations of the two species is indicated by a circle in which the sex chromosome (number in the complements) at the top and morphology of chromosome 7 at the bottom are described. White cross-box symbols indicate intra- or interspecific hybridization. “Auto.” indicates autosome.

**Table 1 genes-15-00300-t001:** Number of sex-linked markers isolated from *Rana sakuraii* and *R. tagoi*.

	Population (City)	No. of frogs analyzed			No. of markers heterozygous or present in				
	**Geographic group**				Male		Female		Total
Species	Sex chromosome				SNP	SilicoDArT	SNP	SilicoDArT	
** *R. sakuraii* **	Saitama (Hanno)	Male	9	Initial sex-linked markers	197	17	13	3	
	**East Japan**	Female	10	False positive	8	0	0	0	
	Hetero (No. 7 M/st)			True sex-linked markers	**189**	**17**	**13**	**3**	222
	Gifu (Gifu)	Male	10	Initial sex-linked markers	12	3	17	3	
	**West Japan**	Female	10	False positive	3	0	6	0	
	Homo			True sex-linked markers	**9**	**3**	**11**	**3**	26
** *R. tagoi* **	Chiba (Ichihara)	Male	10	Initial sex-linked markers	72	15	36	4	
	**East Japan**	Female	10	False positive	7	0	4	0	
	Hetero (No. 7 St/st)			True sex-linked markers	**65**	**15**	**32**	**4**	116
	Nagasaki (Nagasaki)	Male	9	Initial sex-linked markers	16	5	3	3	
	**South Japan**	Female	10	False positive	4	0	0	0	
	Hetero (No. 13)			True sex-linked markers	**12**	**5**	**3**	**3**	23
	Akita-A (Higashi-naruse)	Male	8	Initial sex-linked markers	21	6	19	2	
	**North Japan**	Female	8	False positive	14	0	15	0	
	Homo			True sex-linked markers	**7**	**6**	**4**	**2**	19
	Akita-B’ (Higashi-naruse)	Male	7	Initial sex-linked markers	421	253	106	42	
	**North Japan**	Female	7	False positive	22	0	33	0	
	Hetero (No. 7 M/st)			True sex-linked markers	**399**	**253**	**73**	**42**	767

The sex-linked markers of *Rana sakuraii* and *R. tagoi* were isolated from 146,383 (SNP) and 46,161 (SilicoDArT) markers.

**Table 2 genes-15-00300-t002:** Sex chromosomes identified in *R. tagoi* and *R. sakuraii*.

Species	Population	City	Geographic Group	Sex chr.	Morphology ^(1)^	Chromosome 7 ^(2)^	Analysis	Reference ^(3)^
*R. tagoi*	Akita A	Higashi-naruse	North Japan	**3**	Homo	st/st	SNP	This study
	Akita-B	Higashi-naruse		?	Homo	M/M	Chromosome	This study
	Akita-B’	Higashi-naruse		**7**	Hetero	**M/st**	SNP/Chromosome	This study
	Chiba	Ichihara	East Japan	**7**	Hetero	ST/st	SNP/Chromosome	This study
	Tokyo	Akiruno		**7**	Hetero	ST/st	Chromosome	1
	Ehime	Kumankogen	South Japan	**13**	Hetero	M/M	C-banding	2
	Nagasaki	Nagasaki		**13**	Hetero	M/M	SNP/C-banding	2, this study
*R. sakuraii*	Tokyo, Saitama	Akiruno, Hanno	East Japan	**7**	Hetero	**M/st**	SNP/Chromosome	This study, 1, 2
	Gifu	Gifu	West Japan	**13**	Homo	M/M	SNP	This study

^(1)^ Homo, Homomorphy; Hetero, Heteromorphy. ^(2)^ M, metacentric; ST and st, subtelocentic: short arm of ST is a little longer than that of st. ^(3)^ 1, Ryuzaki et al. (1999) [26]; 2, Kuwana et al. (2021) [25].

## Data Availability

The assembly of *R. temporaria* genomic data is available at https://github.com/DanJeffries/Rana-temporaria-genome-assembly/wiki, accessed on 24 May 2021 (Jeffries et al., 2018) [2]. The genetic data of *R. tagoi* and *R.* sakuraii, other than the Appendix A presented in this study, are available upon request from the corresponding author.

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
