# Peer review of "Multiple Transitions between Y Chromosome and Autosome in Tago’s Brown Frog Species Complex"

_genes, 2024, doi:10.3390/genes15030300_

Round 1

Reviewer 1 Report

Comments and Suggestions for Authors

In the study, Ikuo Miura et al. reported multiple transitions between the Y chromosome and autosome in the Japanese Tago’s brown frog complex. Chromosome banding and SNP molecular analyses revealed three distinct sex chromosomes across various populations of these species in Japan. They discovered that these turnovers, particularly involving chromosome 7, were facilitated by hybridization between various populations, with limited subsequent spread of the new chromosome configuration. These novel findings within vertebrates suggest that hybridization plays a key role in the evolution of sex chromosomes and may be crucial in the speciation of these frogs. The paper presents a well-structured thought process, with detailed result data, appropriate analysis, and a comprehensive discussion, offering significant scientific value. Although there are some deficiencies in the writing, the paper can be accepted after revisions.

L21: The abstract lacks sufficient detail; it does not adequately summarize the main findings and conclusions of the paper, and further supplementation is necessary.

L78: The logical flow within the Materials and Methods section of the paper is unclear. The authors should further clarify the analytical thought process and logic of this section, especially in relation to the analysis of the results.

L79: The size of each geographical group studied varies; the paper should clearly describe the principles that guided the sampling process.

L130: The rationale for choosing 80% as a marker for sex linkage should be supported by citations from the literature.

Comments on the Quality of English Language

English language fine. 

Author Response

Dear reviewer,

Thank you for the reviewer’s comments. They were very helpful for revising our manuscript. We answered to all the questions and comments, which are written in blue below, and the revised parts in the text are written in red.

Comments and Suggestions for Authors

In the study, Ikuo Miura et al. reported multiple transitions between the Y chromosome and autosome in the Japanese Tago’s brown frog complex. Chromosome banding and SNP molecular analyses revealed three distinct sex chromosomes across various populations of these species in Japan. They discovered that these turnovers, particularly involving chromosome 7, were facilitated by hybridization between various populations, with limited subsequent spread of the new chromosome configuration. These novel findings within vertebrates suggest that hybridization plays a key role in the evolution of sex chromosomes and may be crucial in the speciation of these frogs. The paper presents a well-structured thought process, with detailed result data, appropriate analysis, and a comprehensive discussion, offering significant scientific value. Although there are some deficiencies in the writing, the paper can be accepted after revisions.

L21: The abstract lacks sufficient detail; it does not adequately summarize the main findings and conclusions of the paper, and further supplementation is necessary.

We put all results of this study in the abstract: identification of sex chromosomes, sex chromosome turnover and occurrence of hybridization between species or geographic populations, followed by its biological meaning.

To make it more informative, we added and revised the following sentences in L25-29 (revised text).

“, we investigated the frogs of geographic populations ranging from northern to southern Japan of two species, Rana tagoi and R. sakuraii (2n = 26). Particularly, the Chiba populations of East Japan and Akita populations of North Japan in Rana tagoi have been for the first time investigated here. As a result, we identified three different sex chromosomes, namely chromosomes 3, 7, and 13 in the populations of the two species.”

L78: The logical flow within the Materials and Methods section of the paper is unclear. The authors should further clarify the analytical thought process and logic of this section, especially in relation to the analysis of the results.

We changed the order of sub-sections in methods: chromosome analysis was placed in 2.2 followed by mitochondrial analyses in 2.3.

L79: The size of each geographical group studied varies; the paper should clearly describe the principles that guided the sampling process.

We added the following sentences to 2.1 of M & M section in L83-91 according to the order of sub-sections in Results.

“2.1. Frogs

The frogs from Chiba populations of East Japan and Akita populations of North Japan in R. tagoi were newly investigated in addition to those used in our previous study [25]. Chiba population was chosen because it is suggested to be ancestral lineage of Tokyo populations in Rana tagoi [18], while Akita population was chosen because we preliminarily found unusual and different karyotype from other populations. In comparison to one or two populations collected from South Japan of R. tagoi and West Japan of R. sakuraii, much more populations of the two species were collected from East Japan in order to deeply investigate the relationship between speciation and sex chromosome evolution in the region.

L130: The rationale for choosing 80% as a marker for sex linkage should be supported by citations from the literature.

We thank the reviewer for the comment. We have not encountered any published account assessing the impact of the concordance threshold on sex-linked loci analyses. We added the rationale of selecting an 80% concordance threshold, including citation in the method section [Lines 141 to 143]

" To incorporate potential recombination between sex chromosomes and possible sex reversal within these species and minor sequencing errors within the data set, we applied an 80% concordance threshold to isolate sex-linked loci [38].”

Reviewer 2 Report

Comments and Suggestions for Authors

The authors have analyzed the chromosome introgression between two sister frog species by examining sex-linked markers of single nucleotide polymorphisms. This is a piece of interesting work. However, there are some major problems needed being verified.  

1. The authors should validate some of the sex-liked SNPs by PCR or qPCR.

2. The authors should provide more details on the method of Karyotypeing frogs, how many metaphases of nuclear they observed in total.

3. The authors should give explanation why they cloned and sequenced the fragment of mitochondrial NADH dehydrogenase, subunit 1 (ND1), but not cytob gene.   

Comments on the Quality of English Language

The English writing is good. 

Author Response

Dear reviewer,

Thank you for the reviewer’s comments. They were very helpful for revising our manuscript. We answered to all the questions and comments, which are written in blue below, and the revised parts in the text are written in red.

The authors have analyzed the chromosome introgression between two sister frog species by examining sex-linked markers of single nucleotide polymorphisms. This is a piece of interesting work. However, there are some major problems needed being verified.  

  1. The authors should validate some of the sex-liked SNPs by PCR or qPCR.

As for validation by PCR, we have two reasons why we did not or could not. One is that the chromosomes of R. temporaria to which sex-linked markers were aligned perfectly coincided with the heteromorphic sex chromosomes of R. tagoi and sakuraii. Figure3 a, b, c and d are those. Therefore, without validation, the sex-linkage is confirmed to be true. In addition, R. sakuraii of West-Japan (Gife)(Fig3f) shows similar alignment result to that of R. tagoi from South Japan (Nagasaki)(Fig.3d) which has actually heteromorphic sex chromosome 13 (Kuwana et al., 2021 [25]). Therefore, we judged that their sex chromosomes of Fig3e and d are chromosomes 13. One little uncertain result is Fig.3e. So, we mentioned the necessity of further analysis to confirm the sex-linkage to chromosome 3 in L290-293 in Discussion section of the text. One plausible thing is that in Fig3e, no markers were isolated from chromosome 7 or 13, suggesting that the chromosome 7 of R. tagoi from Akita-A population is not sex chromosome but an autosome. The other reason not for allowing us the validation is that the sex-linked SNP sequences are very short, around 70 bp, and hard to identify the adequate primer sequences. We will be making a next plan to confirm the sex chromosome 3 by genomic sequencing.

  1. The authors should provide more details on the method of Karyotypeing frogs, how many metaphases of nuclear they observed in total.

The number of specimens used for chromosome analyses are shown in Tables S3 and S4. We photographed 10-30 C- or LR-banded metaphases from each frog and chose the best one to show in the figures. These explanations were added to M & M (line 120-122 in the text) as follows.

“The number of specimens used for chromosome analyses are shown in Tables S3 and S4. We photographed 10-30 good C- or LR-banded metaphases from those of each frog that we observed, and presented the best ones in the figures.”

  1. The authors should give explanation why they cloned and sequenced the fragment of mitochondrial NADH dehydrogenase, subunit 1 (ND1), but not cytob gene.   

The main reason is that the previous studies by Eto et al. [20, 21] determined the sequences of  ND1+16S rRNA genes to elucidate the phylogenetic history of geographic populations covering whole Japan. So, using the determined sequence data, we needed to know the phylogenetic positions of Akita-A and B and Chiba populations, which are new one and not included in the previous studies, in the phylogeny of Rana tagoi and R. sakuraii. The accession numbers of sequences from the previous studies used here are shown in Table S2.

We added the following part to M & M in line 100:

“Following the previous phylogenetic studies on Rana tagoi and Rana sakuraii [15, 16], …”

Reviewer 3 Report

Comments and Suggestions for Authors

The study focuses on multiple transitions between the Y chromosome and autosome in two species of brown frog: Rana tagoi and R. sakuraii, found in Japan. The conducted research provides a lot of information in the field of genetics, including cytogenetics of these species. Weak geographic barriers did not contribute to reproductive speciation because both species can hybridize. Additionally, there is no difference in the number of chromosomes in the karyotype (2n = 26) between these two species, which also contributes to free inter- and intraspecific hybridization in brown frogs. The research presented in this study provides a lot of information confirming this hypothesis.

The authors applied appropriate genetic techniques, which yielded very interesting research results. However, there are a couple of things in the paper that could be improved. When performing chromosome staining techniques, the authors refer to publication [45] (lines 185-187), where I could not find either the C-banding procedure or the second technique applied. Please specify which procedure was used for C-banding detection. Also, I do not know which technique was used to detect late-replicating bands. GTG? The authors write about the presence of subtelocentric chromosomes in the seventh pair of brown frog karyotypes Rana tagoi and R. sakuraii. In my opinion, this chromosome pair is subacrocentric because the p arm of the chromosome is clearly visible. In subtelocentric chromosomes, the p arms of the chromosomes are usually hardly visible. Therefore, it is a pity that the authors did not calculate the arm index or centromeric index, which would determine the affiliation of these chromosomes to the appropriate morphological group.

I also have reservations about the quality of Figure 3 photographs. Figure 3a and b are of poor quality. Chromosomes stained with the C-banding technique are very faint, making objective assessment difficult, while Figure 3c and d are too dark, causing the A-T bands to merge in some places into one large block. Please replace the photographs in Figure 3 with higher resolution ones.

The remaining results seem to have been carried out very meticulously. The researchers traced the evolution of sex chromosomes, providing many valuable pieces of information worthy of publication. Therefore, after incorporating the suggested comments, I recommend considering this work for publication.

Author Response

Dear reviewer,

Thank you for the reviewer’s comments. They were very helpful for revising our manuscript. We answered to all the questions and comments, which are written in blue below, and the revised parts in the text are written in red.

The study focuses on multiple transitions between the Y chromosome and autosome in two species of brown frog: Rana tagoi and R. sakuraii, found in Japan. The conducted research provides a lot of information in the field of genetics, including cytogenetics of these species. Weak geographic barriers did not contribute to reproductive speciation because both species can hybridize. Additionally, there is no difference in the number of chromosomes in the karyotype (2n = 26) between these two species, which also contributes to free inter- and intraspecific hybridization in brown frogs. The research presented in this study provides a lot of information confirming this hypothesis.

The authors applied appropriate genetic techniques, which yielded very interesting research results. However, there are a couple of things in the paper that could be improved.

When performing chromosome staining techniques, the authors refer to publication [45] (lines 185-187), where I could not find either the C-banding procedure or the second technique applied. Please specify which procedure was used for C-banding detection. Also, I do not know which technique was used to detect late-replicating bands. GTG?

Thank you for the comment.

This is our mistake for citing the reference. The right reference numbers are 25, 31 and 32. In 25 and 32 the late replication banding is described in detail:using BrdU for corporation instead of thymine to detect late replicated chromosomal regions. C-banding basically followed the original method of Sumner (1972) [31]. We revised [45] to [25, 31 and 32] in L120and 122 in the text.

The authors write about the presence of subtelocentric chromosomes in the seventh pair of brown frog karyotypes Rana tagoi and R. sakuraii. In my opinion, this chromosome pair is subacrocentric because the p arm of the chromosome is clearly visible. In subtelocentric chromosomes, the p arms of the chromosomes are usually hardly visible. Therefore, it is a pity that the authors did not calculate the arm index or centromeric index, which would determine the affiliation of these chromosomes to the appropriate morphological group.

We used the definition of chromosome morphology as follows:

Metacentric, p/p+q = 0.5~0.375

Submetacentric, p/p+q = 0.374-0.25

Subtelocentric, p/p+q = 0.249-0.125

Telocentic, p/p+q = 0.124-0.

‘Subtelocentric’ is synonym to ‘acrocentric’. Here we did not use this word because it is now specific and hard to imagine the morphology.

I also have reservations about the quality of Figure 3 photographs. Figure 3a and b are of poor quality. Chromosomes stained with the C-banding technique are very faint, making objective assessment difficult, while Figure 3c and d are too dark, causing the A-T bands to merge in some places into one large block. Please replace the photographs in Figure 3 with higher resolution ones.

These chromosome figures are the best ones we chose from karyotypes that we observed and photographed.

The centromeric heterochromatin of the chromosomes in Fig.3a and b are definitely faint compared to those of Figure 2a and b. These do not reflect the poor quality of banding pictures but the chromosome characteristics in the population (Akita-A). Please see the staining depth of euchromatin. They are not different from each other between Figs 2 and 3.

The darker regions of LR-banded chromosomes are overlapped regions between chromosomes. We could not find any other better figures having no overlapped chromosomes. The point of these karyotypes we focused on is to show the morphology and banding pattern of chromosomes 7, which are not overlapped with any other chromosomes and showing clearly the characteristics of morphology and banding patterns.

The remaining results seem to have been carried out very meticulously. The researchers traced the evolution of sex chromosomes, providing many valuable pieces of information worthy of publication. Therefore, after incorporating the suggested comments, I recommend considering this work for publication.

Round 2

Reviewer 2 Report

Comments and Suggestions for Authors

The authors have given responses to almost all of questions raised by reviewers and the manuscript is now acceptable as a publication in the Journal.

Reviewer 3 Report

Comments and Suggestions for Authors

Most of my comments were taken into account, but photo 3 was not changed to a better resolution. I have no comments on the stained bands on the chromosomes, only on the quality of the photo. I think this can be improved in an image analysis program that is able to apply darkening and brightening filters depending on the need. Unfortunately, it seems to me that the Authors filtered the chromosomes with replication bands too much - darkening them, and in turn too little filtered the chromosomes with c bands. This is an action that can be easily corrected.

Author Response

Dear Reviewer,

Thank you for the comments.

According to the comments, we tried to improve the Figure 3a and 3b by changing the contrast, where C-banded karyotypes were filtered more to hgher contast, while LR-banded ones were pooly filtered to lower contrast. The revised Figure 3 was inserted in the text.